# Paralemnolins X and Y, New Antimicrobial Sesquiterpenoids from the Soft Coral *Paralemnalia thyrsoide*

**DOI:** 10.3390/antibiotics10101158

**Published:** 2021-09-24

**Authors:** Abdelsamed I. Elshamy, Tarik A. Mohamed, Eman M. Elkady, Ibrahim A. Saleh, Ahmed A. El-Beih, Montaser A. Alhammady, Shinji Ohta, Akemi Umeyama, Paul W. Paré, Mohamed-Elamir F. Hegazy

**Affiliations:** 1Department of Natural Compounds Chemistry, National Research Centre, 33 El-Bohouth St., Dokki, Giza 12622, Egypt; ai.el-shamy@nrc.sci.eg; 2Chemistry of Medicinal Plants Department, National Research Centre, 33 El-Bohouth St., Dokki, Giza 12622, Egypt; ta.mourad@nrc.sci.eg (T.A.M.); ia.saleh@nrc.sci.eg (I.A.S.); 3National Institute of Oceanography & Fisheries, NIOF, Cairo 11516, Egypt; emelkady@yahoo.com (E.M.E.); coralreef_niof1@yahoo.com (M.A.A.); 4Chemistry of Natural& Microbial Products Department, National Research Centre, Dokki, Giza 12622, Egypt; aa.el-beih@nrc.sci.eg; 5Graduate School of Integrated Sciences for Life, Hiroshima University, 1-7-1 Kagamiyama, Higashi-Hiroshima 739-8521, Japan; ohta@hiroshima-u.ac.jp; 6Faculty of Pharmaceutical Sciences, Tokushima Bunri University, Yamashiro-cho, Tokushima 770-8514, Japan; umeyama@ph.bunri-u.ac.jp; 7Department of Chemistry & Biochemistry, Texas Tech University, Lubbock, TX 79409, USA; 8Department of Pharmaceutical Biology, Institute of Pharmaceutical and Biomedical Sciences, Johannes Gutenberg University, Staudinger Weg 5, 55128 Mainz, Germany

**Keywords:** *Paralemnalia thyrsoide*, sesquiterpenes, paralemnolins, antimicrobial

## Abstract

The organic extracts of the Red Sea soft coral *Paralemnalia thyrsoides* has led to the identification of two neolemnane-type sesquiterpenoids: paralemnolins X and Y (**1**, **2**). In addition to these newly characterized compounds, ten known metabolites (**3**–**12**) were isolated. Previously reported compounds were elucidated by literature comparison of spectroscopic data (1D and 2D NMR as well as MS data). In vitro antimicrobial activity was investigated for compounds (**1**–**12**) against *Staphylococcus aureus*, *Escherichia coli*, *Candida albicans* and *Aspergillus niger*. Compound **5** showed antimicrobial activity against all assayed microorganisms.

## 1. Introduction

While natural product research includes plant, marine animal and microbe sources, more recently, marine flora and fauna have afforded an exciting number of potential therapeutic agents that are currently in preclinical and clinical evaluation due to promising biological activities with in vivo and in vitro assays [1,2,3,4,5,6,7]. Such drug leads have also significantly contributed to our understanding of cellular biochemical processes. Typical marine skeletons combined with robust biological activities have attracted the attention of pharmacists, chemists and biologists [8,9,10,11].

With the high endemic biota observed in the Red Sea, this marine region serves as an epicenter for marine biodiversity. Indeed, the Red Sea is home to over 40% of the 180 soft coral species that have been discovered worldwide [12]. Soft corals of the genus *Lemnalia* and *Paralemnalia*, in particular, have been discovered to be a significant source of bioactive chemicals such as sesquiterpenoids of the nardosinane, neolemnane and africanane types [13,14,15,16]. Norsesquiterpenes have been found to be abundant in *Paralemnalia thyrsoides* (Alcyonaceae) soft coral [14,16,17,18,19,20,21]. Previous chemical studies of *P. thyrsoides* resulted in the isolation of sesquiterpenoids known as paralemnolins A—W [18,19,21,22,23].

Here we report on two neolemnane-type sesquiterpenoids isolated from the Formosan soft coral *P. thyrsoides.*

## 2. Results and Discussion

Chromatographic fractionation and purification of the EtOAc extract of *P. thyrsoides* afforded two new sesquiterpenes known as paralemnolin X (**1**) and paralemnolin Y (**2**), as well as 10 previously reported compounds, paralemnolin D (**3**) [18], paralemnolin E (**4**) [18], paralemnolin J (**5**) [21], paralemnolin P (**6**) [21], 2-deoxy-12-oxolemnacarnol (**7**) [14], paralemnolide A (**8**) [24], parathyrsoidin G (**9**) [25], 6α-acetyl-1(10)-α-13-nornardosin-7-one (**10**) [15], 6α-acetyl-7α-acetate-1(10)-α-13-nornardosine (**11**) [15], and clavukoellian F (**12**) [26] (Figure 1). All isolated secondary metabolites were elucidated by 1D and 2D-NMR as well as mass spectroscopy (Appendix A).

Compound **1** was obtained in the form of a colourless oil exhibiting a positive optical rotation in methanol ${\left[\alpha \right]}_{D}^{25}$ + 67.5 (*C* 0.01, MeOH). CI-MS analysis showed a molecular ion peak at 336.1940, indicating a molecular formula of C_19_H_28_O_5_ (calculated 336.1939 for C_19_H_28_O_5_) and thus indicating six degrees of unsaturation. The ^13^C and ^1^H NMR spectra revealed the presence of two acetyl groups (δ_H_ 2.08 (3H, s), 2.00 (3H, s); δ_C_169.9 (2C=O), 21.0 (2CH_3_)) and one double bond (δ_C_ 127.1 (CH), 143.7 (C)).The ^13^C NMR (Table 1) and DEPT spectra revealed 19 carbon signals, which were classified as 5 quaternary carbons (comprising two acetate keto groups at δ_C_ 169.9), 5 methines (including three oxygenated methines at δ_C_ 65.3, 72.5, 73.8 and one olefinic methines at δ_C_ 127.1), 4 methylenes, and 5 methyles (including two methyl groups of acetyl groups) in the ^13^C NMR spectrum, indicating the presence of two acetyl groups. Furthermore, one trisubstituted double bond at 127.1 (CH) and 143.7 (C) was assigned from the ^13^C NMR, DEPT and HMQC spectra. The suggested structure, a neolemnanoid skeleton, corresponded to a bicyclic structure with a condensed eight- and six-membered ring system, as previously isolated from *P. thyrsoides* [18]. The skeleton was established by extensive 2D NMR analysis (HMQC, HMBC and ^1^H–^1^H COSY). The ^1^H–^1^H COSY spectrum was used to reveal proton sequences and the connectivity of H-4/H-5, H-5/H_2_-6, H_2_-6/H_2_-7, H-9/H_2_-10, and H_2_-10/H_2_-11 (Figure 2). Two oxygen-bearing carbons at δ_C_ 55.4 (C) and 65.3 (CH), the latter coinciding with a proton at δ_H_ 2.96 (s) in the HMQC spectrum, confirmed a trisubstituted epoxide (Figure 2). Comparison of the NMR spectral data of compounds paralemnolin D, E, G and H revealed that the C-2–C-3 double bond was oxidized to bean epoxide group in compound **1 [18]**. The epoxide site was established via H_3_-14 correlation to carbon signals at δ_C_ 65.3 (C-2), 55.4 (C-3), and 72.5 (C-4) in HMBC spectrum. Spectroscopic data was similar to paralemnolin F [18], except the up-field shift of C-2 at δ_C_65.3/71.8 as well as the down-field shift of H-2 at δ_H_ 2.96 (Table 1). NMR spectral data comparison of compound **1** with paralemnolin F disclosed that both compounds had similar structures. In Table 1, the NOESY spectrum was recorded, and H_3_-13 was found to show NOE interactions with H_3_-14, H_3_-15 and one proton (δ_H_ 1.42, m) of H_2_-11; additionally, H_3_-14 was found to show NOE interactions with H-4 and H-5, identifying these protons as β-oriented. Furthermore, H-2 (δ_H_ 2.96, s) exhibited NOE interactions with H-12, suggesting the α-orientation of H-2 (Figure 3). The structure of **1** was determined as paralemnolin X.

Compound **2** (a colourless oil) displayed a positive optical rotation in methanol ${\left[\alpha \right]}_{D}^{25}$ + 25.6 (C 0.01, MeOH). CI-MS analysis showed a molecular ion peak at 336.1934, indicating a molecular formula of C_19_H_28_O_5_ (calculated 336.1933, for C_19_H_28_O_5_) and thus indicating six degrees of unsaturation. The existence of two acetyl groups (δ_H_ 2.11 (3H, s), 2.02 (3H, s); δ_C_ 169.9, 169.7 (2C=O), 20.9, 20.7 (2CH_3_)) and one double bond (δ_C_ 124.2 (CH), 142.1 (C)) were revealed by the ^13^C and ^1^H NMR spectra. The ^13^C NMR spectrum displayed 19 carbon signals, which were classified as 5 quaternary carbons (comprising two acetate keto groups at δ_C_169.7, 169.9), 5 methines (including three oxygenated methines at δ_C_ 65.2, 75.5, 73.6 and one olefinicmethine at δ_C_ 124.2), 4 methylenes, and 5 methyles (including two methyl groups of acetyl groups) in the ^13^C-NMR spectrum, indicating the presence of two acetyl groups. Furthermore, one trisubstituted double bond at 124.2 (CH) and 142.1 (C) was assigned from the ^13^C-NMR, DEPT and HMQC spectra. These structural elements recommended that the neolemnanoid skeleton confers a bicyclic structure with a condensed 6/8-membered ring system, as previously isolated from *P. thyrsoides* [18]. The skeleton of 2 was established by extensive 2D NMR analysis (^1^H–^1^H COSY, HMQC, and HMBC). To establish the proton sequences, the ^1^H–^1^H COSY spectrum was used to reveal the connectivity of H-4/H-5, H-5/H_2_-6, H_2_-6/H_2_-7, H-9/H_2_-10, and H_2_-10/H_2_-11 (Figure 2). A trisubstituted epoxide was established from two oxygen-bearing carbons at δ_C_ 58.3 (C) and 65.2 (CH). The latter correlated with a proton at δ_H_ 2.89 (s) in the HMQC spectrum. Spectroscopic data was similar to **1,** except for the down-field shift of C-3 at δ_C_ 58.3/55.4 and the up-field shift of C-14 at δ_C_ 13.9/19.5 (Table 1). The NMR spectra, in comparison with 1, revealed that both compounds had similar structures; in Table 1, the NOESY spectrum was recorded. H_3_-13 show NOE interactions with protons at δ_H_ 0.98 d (6.8, H_3_-15), and 2.89 (s, H-2) and one proton at δ_H_ 2.40 (m, H_2_-7); additionally, H-2 shows NOE interactions with protons at δ_H_ 4.77 (d, 6.0, H-4) and 5.09 (dd, 10.0, 6.0, H-5) and one proton at δ_H_1.81 (m, H_2_-6), revealing a β-orientation of these protons. Furthermore, H_3_-14 (δ_H_1.48, s) exhibited NOE interactions with protons at δ_H_1.77 (m, H-12) and one proton at δ_H_ 2.10 (m, H_2_-6), suggesting the α-orientation H_3_-14 (Figure 3). The structure of compound **2** was determined as paralemnolin Y.

A microdilution assay was used to determine the MIC. Metabolite **5** was the most active compound against all test microorganisms, with weak antimicrobial activity in comparison with positive controls (treflucan and thiophenicol) while other compounds showed varying activities (see Table 2). Compound **7** exhibited no observed antimicrobial activity.

## 3. Materials and Methods

### 3.1. General Experimental Procedures

For optical rotation the JASCO P-2300 polarimeter (JASCO, Tokyo, Japan) was used; for IR spectra, the Shimazu FTIR-8400S instrument (Shimazu, Columbia, MD 21046, USA) was used. A Bruker 600 or 500 Hz NMR spectrometer was used to record 1D and 2D NMR spectra (Bruker, MA, USA). A JEOL JMS-700 equipment was used to obtain HR-MS spectra (Tokyo, Japan). TLC analysis was conducted with precoated silica gel plates (Merck, Kieselgel60 F_254_, 0.25 mm, Merck, Darmstadt, Germany); chromatography (CC) was conducted with silica gel 60 (Merck, 230–400 mesh, Merck, Darmstadt, Germany). HPLC was carried out with a Jasco PU-980 pump, a Jasco UV-970 intelligent UV detector at 210 nm, and a semi-preparative reversed-phase column (Cosmosil C_18_, column 250 × 10 mm, 5 μm, Nacalai Tesque, Kyoto, Japan).

### 3.2. Coral Material, Extraction and Separation

A collected Red Sea soft coral *Paralemnalia thyrsoide* (Hurghada in March 2017) was identified by Montaser A. Alhammady (co-author) with a voucher specimen (08RS1075) deposited in the National Institute of Oceanography and Fisheries (NIOF), Egypt.

The soft coral *P. thyrsoides* (2.5 kg wet weight) was sliced into small parts; this was followed by extraction with ethyl acetate (6 L × 3 times). The combined extracts afforded a dark black gum (117.5 g) under vacuum concentration. Further chromatography fractionations and purification were operated using our previously described protocol [27]. All column fractions were examined via TLC and collected as the main fractions (PT1-PT9). One fraction (PT3; 930 mg) was subjected to the silica gel column with an elution solvent system of *n*-hexane/CHCl_3_ step gradient to afford two sub-fractions (PT3-1-2) via the TLC profile. The sub-fraction PT3-1 was re-purified and rephrased. RP-18 HPLC (MeOH/H_2_O, 3:1) afforded compounds 1 (9.1 mg), **2** (13.4 mg), **6** (7.3 mg), **7** (5.8 mg), and **9** (10.5 mg). Fraction PT-4 (718.9 mg) was further fractionated over ODS-C18 CC using MeOH/H_2_O with increasing polarity as the eluent; this afforded three main subfractions (PT4-1-3). The subfraction PT4-2 (278.7 mg) was re-subjected to RP-18 HPLC (MeOH–H_2_O, 4:1) to give compounds 3 (7.9 mg), 4 (10.2 mg), 5 (9.3 mg), and 8 (6.7 mg). The sub-fraction PT4-3 was purified by Isolera flash column chromatography eluted with *n*-hexane:EtOAc (1:1) to afford compounds **10** (8.6 mg), **11** (12.6 mg), and **12** (5.8 mg).

### 3.3. Spectral data of Paralemnolin X, Y (**1**,**2**)

Paralemnolin X (**1**): yellow oil; ${\left[\alpha \right]}_{D}^{25}$ + 67.5 (c 0.01, CH_3_OH); ^1^H and ^13^C NMR data (CDCl_3_, 500 Hz), see Table 1; HRCIMS *m*/*z* 336.1940 (calculated 336.1939 for C_19_H_28_O_5_).

Paralemnolin Y (**2**): yellow oil; ${\left[\alpha \right]}_{D}^{25}$ + 25.6 (c 0.01, CH_3_OH); ^1^H and ^13^C NMR data (CDCl_3_, 500 Hz), see Table 1; HRCIMS *m*/*z* 336.1934 (calculated 336.1933 for C_19_H_28_O_5_).

### 3.4. Antimicrobial Activity Assay

#### 3.4.1. Microorganisms

*Staphylococcus aureus* (ATCC29213) and *Escherichia coli* (ATCC 25922) are Gram-positive bacterium (GPB) and Gram-negative bacterium (GNB), respectively. Additionally, the fungal yeast *Candida albicans* (ATCC 10231) and *Aspergillus niger* (NRRL-599) were screened for compound anti-microbial activity by the Microbial and Natural Products Chemistry Department, National Research Centre (NRC), Egypt.

Nutrient agar medium was used for bacterial cultivation followed by suspension in nutrient broth medium at 37 °C for 24 h. The fungus culture was grown on potato dextrose agar medium at 28 °C for 4 days, and then suspended in potato dextrose broth. The turbidity of the suspension was adjusted to that of the standard 0.5 McFarland solution.

#### 3.4.2. Minimum Inhibitory Concentration Determination

The MIC values were determined by the broth microdilution assay (NCCLS, 2008) [28] with slight modifications. Each sample was initially dissolved in DMSO, and subsequently diluted with broth media to reach the desired final concentration. Five-fold dilutions were prepared in a 96-well plate. The microbial suspensions were added into each well, then incubated at 37 °C for 24 h for bacteria and at 28 °C for 72 h for fungi. The MIC value was determined as the lowest concentration of the sample that inhibited microbial growth. The assay was carried out in nutrient broth medium for bacteria and potato dextrose broth medium for fungi. The assay was performed according to Hammer et al. (1999) [29], with slight modifications. Briefly, 1 mg of the pure compound was dissolved in 50 µL DMSO, and 10 µL was added as the initial concentration in the first column of the sterile polystyrene 96-well plates. Then, 190 µL of the tested microbial suspension adjusted to 5 × 10^5^ CFU/mL was added. Serial dilutions were performed by the addition of 100 µL of the first column to the second one, and so on. The final volume was adjusted to 200 µL on each well by the addition of the microbial suspension to obtain final concentrations of the tested compounds ranging from 100 to 3.125 µg. Microbial growth controls were made by replacing the tested compound with the same volume of DMSO (in order to eliminate the possible antibacterial effect of the solvent). Sterility controls were prepared by using broth media alone. The plates were covered with a sterile plate sealer, carefully mixed and incubated at 37 °C for 24 h for bacteria and 28 °C for fungi. Microbial growth was indicated by the turbidity. The absence of microbial growth was interpreted as antimicrobial activity. The MIC value was taken as the lowest concentration of the test agent that caused complete inhibition (100%) of microbial growth [30].

## 4. Conclusions

A *P. thyrsoide* extract afforded two new neolemnane-type sesquiterpenoids, paralemnolins X, Y (**1, 2**) and ten known secondary metabolites (**3**–**12**). Chemical structures were elucidated based upon spectroscopic analyses. Only compound **5** exhibited antimicrobial activity against all test microorganisms, followed by compound **9.** Other compounds showed varying activities against different test microorganisms.

## Figures and Tables

**Figure 1 antibiotics-10-01158-f001:**
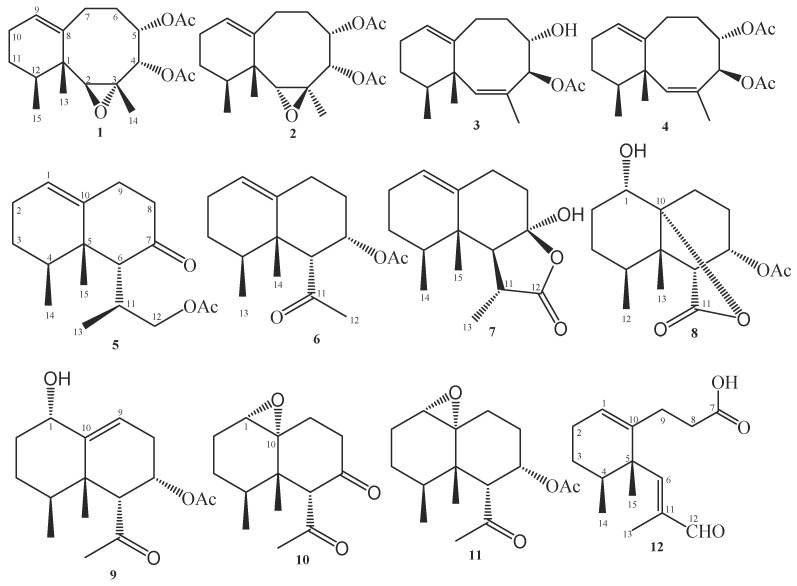
Structures of isolated compounds (**1**–**12**).

**Figure 2 antibiotics-10-01158-f002:**
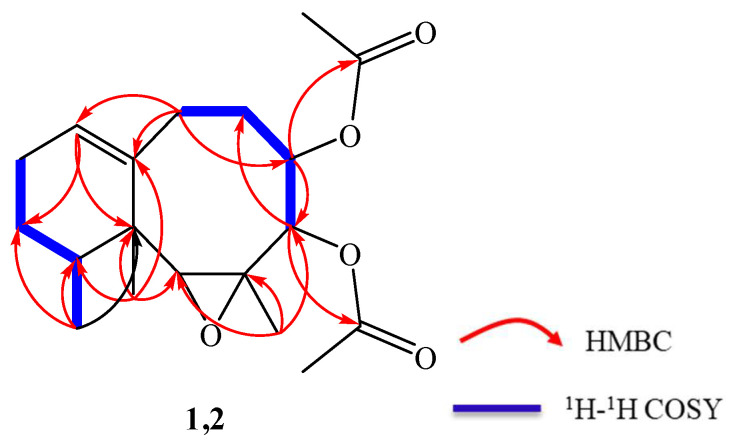
Key HMBC and ^1^H–^1^H COSY of **1, 2**.

**Figure 3 antibiotics-10-01158-f003:**
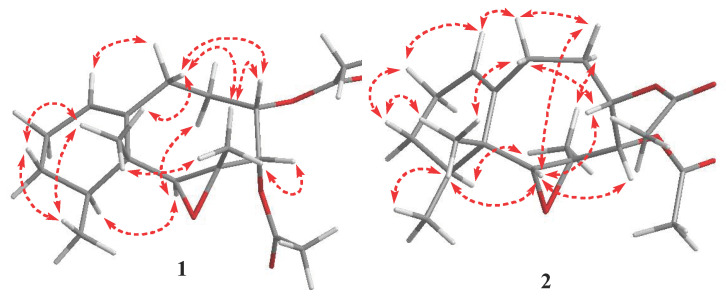
Significant NOESY of paralemnolin X, Y (**1****,2**).

**Table 1 antibiotics-10-01158-t001:** ^1^H (CDCl_3_, 500 MHz) and ^13^C (125 MHz) NMR of **1** and **2**.

No	1		2		Paralemnolin F	
	*δ_H_*	*δ_C_*	*δ_H_*	*δ_C_*	*δ_H_*	*δ_C_*
1	---	42.3	---	38.9	---	40.9
2	2.96 s	65.3	2.89 s	65.2	2.62 s	71.8
3	---	55.4	---	58.3	---	58.7
4	5.18 d (3.0)	72.5	4.77 d (6.0)	75.5	5.30 br s	73.6
5	5.13 dd (11.4, 2.4)	73.8	5.09 dd (10.0, 6.0)	73.6	5.27 m	71.9
6	1.82 m, 1.90 m	31.3	1.81 m, 2.10 m	31.9	1.86 m, 1.97 m	30.2
7	2.05 m, 2.41 m	30.0	2.40 m, 2.44 m	32.3	2.15m, 2.42 m	29.8
8	---	143.7	---	142.1	---	140.4
9	5.55 dd (4.3, 2.6)	127.1	5.44 br d (4.5)	124.2	5.58 dd (3.6, 3.8)	127.8
10	2.07 m	25.9	1.96 m, 2.00 m	25.8	2.07 m	25.3
11	1.42 m, 1.46 m	25.3	1.36 m, 1.44 m	26.3	1.50 m	25.9
12	1.80 m	39.6	1.77 m	32.3	1.73 m	42.8
13	0.80 s	14.5	1.20 s	21.9	0.98 s	17.0
14	1.73 s	19.5	1.48 s	13.9	1.48 s	20.7
15	0.98 d (6.9)	16.9	0.94 d (6.9)	17.7	0.97 d (6.9)	16.2
CH_3_-AC	2.00 s, 2.08 s	21.0	2.11 s, 2.02 s	20.7, 20.9	2.07s, 2.09 s	20.9, 21.1
C=O-Ac	---	169.9	---	169.7, 169.9		170.0, 169.9

**Table 2 antibiotics-10-01158-t002:** Minimal inhibitory concentration (MIC- µmol) determined by microdilution assay.

Compound No.	*Staphylococcus aureus*	*Escherichia coli*	*Candida albicans*	*Aspergillus niger*
**1**	1.488	-	-	-
**2**	1.488	-	-	-
**3**	0.899	-	-	1.798
**4**	1.562	-	-	1.562
**5**	0.449	1.798	1.798	2.248
**6**	1.893	-	-	-
**7**	-	-	-	-
**8**	0.221	-	-	1.773
**9**	1.785	-	1.785	0.446
**10**	2.110	-	-	1.059
**11**	1.785	-	-	0.446
**12**	0.992	-	-	0.992
Treflucan	-	-	0.327	0.163
Thiophenicol	0.140	0.281	-	-

## Data Availability

The data presented in this study are available in Appendix A.

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
