# Peer review of "Paralemnolins X and Y, New Antimicrobial Sesquiterpenoids from the Soft Coral Paralemnalia thyrsoide"

_antibiotics, 2021, doi:10.3390/antibiotics10101158_

Round 1

Reviewer 1 Report

The manuscript describe the isolation and structure elucidation of two new natural products 1-2, and ten known compounds 3-12 from soft coral Paralemnalia thyrsoide collected at Red Sea. Finally, isolated compounds were tested for their ability against microbial strains. 

1) The introduction should be improved, it lacked of fundamental information about the focus sesquiterpenoids, paralemnolins. Paralemnolins A-W should be included in introduction.

2) There are too many self-citation.

3) There is no numbering system in Fig. 1.

4) It seem that 1 (9.1 mg) and 2 (13.4 mg) were afforded in good amount of yield. The supplementary materials showed that 1 and 2 were not really in pure form, obtaining a pure form for bioassay and providing a clear NMR spectra for future reference is very important for this manuscript as 1-2 are reported as a new natural products.

5) How is the antibacterial assay was carried out? Usually we use two time dilution starting with 250, 125, 64, 32, 16, 8, 4, 2 uM. The values in Table 2 is really unbelievable.

6) The identity of expert that identify studied soft coral should be included in manuscript, for transparency.  

Author Response

The introduction should be improved, it lacked of fundamental information about the focus sesquiterpenoids, paralemnolins. Paralemnolins A-W should be included in introduction.

A statement about sesquiterpenes (Paralemolins A-W) have been added to introduction with cited references (line 50-51): "Previous chemical studies of P. thyrsoide resulted in the isolation of sesquiterpenoids, paralemnolins A—W [18,19,21-23]."

There are too many self-citation.

Self-citations 6 and 7 were removed; other citations related to Red Sea studies are highly relevant to the study. Removed references: 

Ibrahim, M.A.A.; Abdelrahman, A.H.M.; Atia, M.A.M.; Mohamed, T.A.; Moustafa, M.F.; Hakami, A.R.; Khalifa, S.A.M.; Alhumaydhi, F.A.; Alrumaihi, F.; Abidi, S.H. Blue Biotechnology: Computational Screening of Sarcophyton Cembranoid Diterpenes for SARS-CoV-2 Main Protease Inhibition. Marine drugs 2021, 19, 391.

Ibrahim, M.A.A.; Abdelrahman, A.H.M.; Hussien, T.A.; Badr, E.A.A.; Mohamed, T.A.; El-Seedi, H.R.; Pare, P.W.; Efferth, T.; Hegazy, M.-E.F. In silico drug discovery of major metabolites from spices as SARS-CoV-2 main protease inhibitors. Computers in Biology and Medicine 2020, 126, 104046.

There is no numbering system in Fig. 1.

Structure numbering has been added to Figure 1.

4) It seem that (9.1 mg) and 2 (13.4 mg) were afforded in good amount of yield. The supplementary materials showed that and 2 were not really in pure form, obtaining a pure form for bioassay and providing a clear NMR spectra for future reference is very important for this manuscript as 1-2 are reported as new natural products.

At the beginning of the chemical characterization process, it was thought that the isolated metabolites 1-2 were known compounds (especially since several known compounds from the A-W previously reported compounds were isolated). As such, a low number of scans were run for NMR analysis, followed by biological activity. Through the purification/ identification/bioassay process, it was established that 1-2 were in fact new compounds but at that point it was not possible to re-submit samples for additional NMR analysis. 

How is the antibacterial assay was carried out? Usually we use two time dilution starting with 250, 125, 64, 32, 16, 8, 4, 2 uM. The values in Table 2 are really unbelievable.

The following information was added to the Materials and Methods section (lines 206-229): "The MIC values were determined by the broth microdilution assay (NCCLS, 2008) [26] with slight modification. Each sample was initially dissolved in DMSO, and subsequently diluted with broth media to reach the desired final concentration. Five-fold dilutions were prepared in a 96-well plate. The microbial suspensions were added into each well and then incubated at 37º C for 24 h for bacteria and at 28 ºC for 72 h for fungus. The MIC value was determined as the lowest concentration of sample that inhibited the microbial growth. The assay was carried out in nutrient broth medium for bacteria and potato dextrose broth medium for fungus. The assay was performed according to Hammer et al. (1999) [27], with slight modifications. Briefly, 1 mg of the pure compound was dissolved in 50 µl DMSO and 10 µl was added as initial concentration in the first column of the sterile polystyrene 96 well plates. Then 190 µl of the tested microbial suspension adjusted to 5 x 105 CFU / ml was added. Serial dilutions were done by addition of 100 µl of first column to second one and so on. The final volume was adjusted to 200 µl on each well by addition of the microbial suspension to get final concentrations of tested compounds from 100 to 3.125 µg. Microbial growth controls were made by replacing the tested compound with the same volume of DMSO (in order to eliminate the possible antibacterial effect of the solvent). Sterility controls were prepared by using broth media alone. The plates were covered with a sterile plate sealer, carefully mixed and incubated at 37 ºC for 24 h for bacteria and 28 ºC for fungus. Microbial growth was indicated by the turbidity. The absence of microbial growth was interpreted as antimicrobial activity. The MIC value was taken as the lowest concentration of the test agent that caused complete inhibition (100%) of microbial growth [28].

The values reported in the table were calculated in uM as we have pure compounds with known molecular weights.

The identity of expert that identify studied soft coral should be included in manuscript, for transparency.  

The person who identify the soft coral is reported (line 168): "The collected Red Sea soft coral Paralemnalia thyrsoide (Hurghada in March 2017) was identified by Montaser A. Alhammady (co-author) with a voucher specimen (08RS1075) deposited in the National Institute of Oceanography and Fisheries (NIOF), Egypt."

Reviewer 2 Report

The work is well argued especially regarding the spectroscopy characterization of all isolated compounds. However some inaccuracy about compound 1 (paralemnolins X) stands out: ref18 (Chem. Pharm. Bull. 2007) reports the compound, with the same stereochemistry, indicating the beta orientation of H2 proton while in this work it is attributed an alpha orientation to H-2 proton through the NOE analysis (row 96: 'H-2 exhibited NOE interaction with H-2 suggesting the alpha orientation of H-2). If compound 1 is a new compound it should have a different stereochemistry from the one indicated in ref 18 and it has to be clearly indicated in Fig. 1

Please clarify this point.

In row 73: instead of δH 127.1 should be corrected in δC 127.1

Row 87: the work mentions Paralemnolin H which is not described in fig.1

Row 108: instead of δH 124.2 should be corrected in δC 124.2

In table 1 it should be indicated which solvent has been used and at which MHz spectra have been recorded and that in parentheses J values (in Hz) are indicated.

In conclusions section: It should be highlighted which compounds are effective as antimicrobial and against which microorganisms because, as it has written in row 144, not all compounds showed activity against all studied microorganisms.

References section: as it is suggested in author’s instructions, abbreviated journal name should be used.

Author Response

In row 73: instead of δH1 should be corrected in δC127.1.

Line 73 the chemical shift was corrected to 127.1.

Row 87: the work mentions Paralemnolin H which is not described in fig.1

Paralemnolin H is not an isolated compound but is a literature-reported metabolite.

Row 108: instead of δH2 should be corrected in δC124.2

 Typo has been corrected.

In table 1 it should be indicated which solvent has been used and at which MHz spectra have been recorded and that in parentheses values (in Hz) are indicated.

The solvent that was used has been added.

In conclusions section: It should be highlighted which compounds are effective as antimicrobial and against which microorganisms because, as it has written in row 144, not all compounds showed activity against all studied microorganisms.

The conclusion has been revised with the added statement (lines 233-236): "Only compound 5 exhibited antimicrobial activity against all test microorganisms followed by compound 9. Other compounds showed varying activities against different test microorganisms."

6- References section: as it is suggested in author’s instructions, abbreviated journal name should be used.

The endnote style for MDPI was used (https://endnote.com/style_download/mdpi/) MDPI wrote in the MDPI Reference List and Citations Style Guide

Note: If you are not sure how to abbreviate a particular journal title, please leave the entire title. The Editorial Office will abbreviate those journal titles appropriately.”

Round 2

Reviewer 1 Report

Dear Authors,

The major concerns have been addressed positively, there is no further comments from me.